# Complex charge density waves in simple electronic systems of two-dimensional III$_2$−VI$_3$ materials

Yu-Ting Huang [1], Zhen-Ze Li [1,2], Nian-Ke Chen [1] ✉, Yeliang Wang [3], Hong-Bo Sun [1,2], Shengbai Zhang[4] & Xian-Bin Li [1] ✉

Charge density wave (CDW) is the phenomenon of a material that undergoes a spontaneous lattice distortion and modulation of the electron density. Typically, the formation of CDW is attributed to Fermi surface nesting or electron-phonon coupling, where the CDW vector ($\boldsymbol{Q}_{CDW}$) corresponds to localized extreme points of electronic susceptibility or imaginary phonon frequencies. Here, we propose a new family of multiple CDW orders, including chiral Star-of-David configuration in nine 2D III$_2$−VI$_3$ van der Waals materials, backed by first-principles calculations. The distinct feature of this system is the presence of large and flat imaginary frequencies in the optical phonon branch across the Brillouin zone, which facilitates the formation of the diverse CDW phases. The electronic structures of 2D III$_2$−VI$_3$ materials are relatively simple, with only III-$s,p$ and VI-$p$ orbitals contributing to the formation of the CDW order. Despite that, the CDW transitions involve both metal-to-insulator and insulator-to-insulator transitions, accompanied by a significant increase in the bandgap caused by an enhanced electronic localization. Our study not only reveals a new dimension in the family of 2D CDWs, but is also expected to offer deeper insights into the origins of the CDWs.

The concept of charge density wave (CDW) refers to the phenomena of periodic modulation of solid lattice and charge density[1–3]. The formation of CDW is often accompanied by other interesting physical properties such as ferroelectric[4,5], superconductivity[6–10], magnetism[11–13], and Mott insulation[14–16], thus attracting extensive attention in the field of condensed matter physics. Typically, CDWs can be divided into three categories according to their origin[17]. Type I CDWs often occur in quasi-one-dimensional systems and can be regarded as an analog of the Peierls instability. Lattice vibration will be effectively screened by the zero electronic excitation at the Fermi surface nesting (FSN) vector, and a dimerized, metal-to-insulator transition occurs, lowering the energy of the ground state[18–23]. However, the simple physical picture of

FSN is difficult to generalize to two-dimensional (2D) or three-dimensional physical systems. In this case, the $\boldsymbol{k}$-dependent electron-phonon coupling (EPC) plays a crucial role, which leads to the emergence of the concept of type II CDWs. The typical feature of such a CDW system is the coincidence of $\boldsymbol{Q}_{CDW}$ with the peak of the phonon linewidth, which corresponds to the localized imaginary frequencies of phonon branches that trigger the CDW structural transition. Most 2D CDW materials such as 2$H$-NbSe$_2$, and 1$T$-VSe$_2$[24–26], can be classified as Type II CDWs, and the instability of electron and lattice structures in these materials is closely correlated. There still exist unconventional Type III CDW materials such as cuprates. Although the possible presence of FSN or EPC in these materials, there is no clear evidence

---

[1]State Key Laboratory of Integrated Optoelectronics, College of Electronic Science and Engineering, Jilin University, Changchun, China. [2]State Key Laboratory of Precision Measurement Technology and Instruments, Department of Precision Instrument, Tsinghua University, Beijing, China. [3]School of Integrated Circuits and Electronics, MIIT Key Laboratory for Low-Dimensional Quantum Structure and Devices, Beijing Institute of Technology, Beijing, China. [4]Department of Physics, Applied Physics, and Astronomy, Rensselaer Polytechnic Institute, Troy, NY, USA. ✉e-mail: chennianke@jlu.edu.cn; lixianbin@jlu.edu.cn

indicating that these factors explicitly determine the formation of specific CDW configurations, whereas the strong electron correlation effect may play a role[27]. Since the origin of CDW remains controversial as its properties are highly material-dependent, exploring unique CDW is of significant importance for understanding their property and origin.

In this work, we unveil unexpected multiple CDW orders in 2D III$_2$−VI$_3$ (III = Al, Ga, In; VI = S, Se, Te) materials, represented by the chiral Star-of-David (c-SoD) configuration of 2D In$_2$Se$_3$, backed by first-principles calculations. Compared with traditional SoD CDW materials such as 1$T$-TaS$_2$, which exhibits strong electronic correlation[28–31], the electronic structures of 2D III$_2$−VI$_3$ materials are quite simple, with only $s$- and $p$-orbitals observable near the Fermi level. We found it is the flat optical phonon branch with significant imaginary frequencies [induced by the Mexican-hat potential energy surface (PES)] distributed over a considerable range of the Brillouin zone that facilitates the formation of a large number of stable/metastable CDW phases. Calculations show that the energies of these CDW phases are close to each other, and in contrast to the widely-studied type II CDWs, their $Q_{CDW}$ do not always correspond to the local minimum of the phonon branch, implying that the EPC is not the sole factor determining the specific CDW configuration. Through electronic band structure analyses, we found that the CDW transitions in 2D III$_2$−VI$_3$ materials are

accompanied by an enhancement of electron localization and a more significant bandgap opening compared to other 2D CDWs. This suggests that the observed CDW in 2D III$_2$−VI$_3$ materials may originate from the "locking" of specific chemical bonding provided by the broad freedom of the Mexican-hat PES. Our discovery of complex CDWs induced by flat phonon branch with significant imaginary frequencies in simple electronic systems is expected to provide deeper insights into the origins of CDWs.

## Results

### c-SoD CDW in 2D β-In$_2$Se$_3$

2D In$_2$Se$_3$ has attracted widespread attention in recent years due to its robust ferroelectricity at the ultrathin thickness limit[32–34]. One of the most important features of 2D In$_2$Se$_3$ is the diversity of phases, such as α, β, γ phases, etc[34]. Figure 1a shows the schematic diagram of the atomic structure of undistorted high-symmetry monolayer β-In$_2$Se$_3$ (β$_c$-In$_2$Se$_3$), where the middle layer Se [Se($m$)] atoms are located at the center of each unit cell. It has a five-atomic-sublayer structure arranged in the order of Se-In-Se-In-Se and exhibits an inversion symmetry centered on Se($m$) atoms. Previous calculations have demonstrated the energetic instability of 2D β$_c$-In$_2$Se$_3$, consequently, the Se($m$) atoms will deviate from the central position along a unified direction, leading to an in-plane ferroelectric polarization[34,35]. Surprisingly, we found that

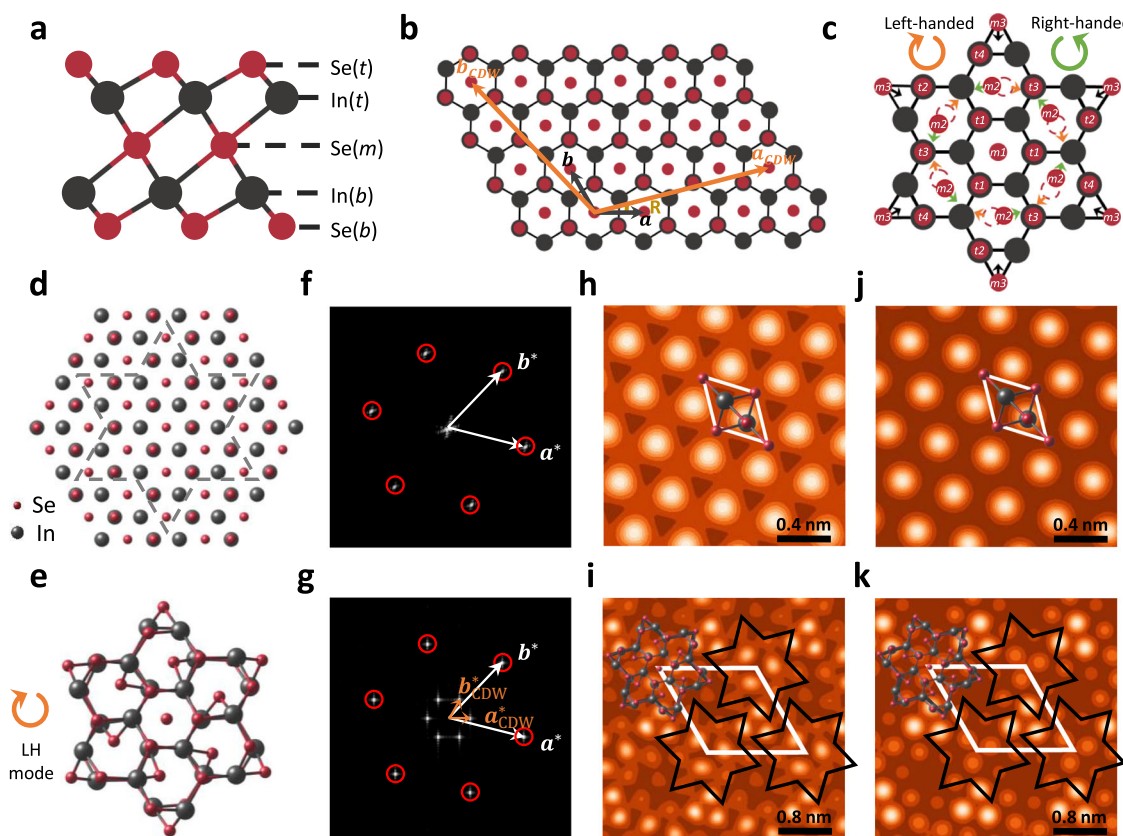

**Fig. 1 | Atomic structures and simulated scanning tunneling microscope (STM) patterns of the pristine high-symmetry phase and chiral Star-of-David (c-SoD) charge density wave (CDW) phase of monolayer β-In$_2$Se$_3$. a** Schematic side view of the atomic structure of the pristine high-symmetry monolayer β$_c$-In$_2$Se$_3$ (β$_c$-In$_2$Se$_3$). **b** Top view of the lattice vectors of monolayer β$_c$-In$_2$Se$_3$ (black arrows) and the c-SoD reconstructed phase (orange arrows), the rotation angle between the two sets of lattices is $R$ = 13.9°. **c** Schematic diagram of the displacement of the middle layer Se [Se($m$)] atoms during the β$_c$-to-c-SoD CDW transition, where Se($m$) atoms are marked as Se($m1$), Se($m2$), Se($m3$) and Se($t$) atoms are marked as Se($t1$), Se($t2$), Se($t3$), Se($t4$) according to their locations. The CDW transition can be named the left-handed (LH) mode and right-handed (RH) mode according to the clockwise or

counterclockwise motions of Se($m2$) atoms. Top view of the actual atomic structures of (**d**) β$_c$ and (**e**) LH-c-SoD phases of monolayer In$_2$Se$_3$. The c-SoD configuration is formed by the atoms enclosed by the gray dotted line in Fig. 1d being twisted together according to the LH mode. **f**, **g** Diagram of the reciprocal lattice vectors extracted from the fast Fourier-transform mapping of the atomic structures of the β$_c$ and c-SoD phases of monolayer In$_2$Se$_3$. White and orange arrows represent the reciprocal lattice vectors of β$_c$ ($a^*$ and $b^*$) and c-SoD ($a^*_{CDW}$ and $b^*_{CDW}$) phase of 2D In$_2$Se$_3$, respectively. Density functional theory simulated STM images of the (**h**, **i**) empty and (**j**, **k**) filled states of β$_c$ and LH-c-SoD phases, the simulated bias voltages are +2 V and −2 V, respectively.

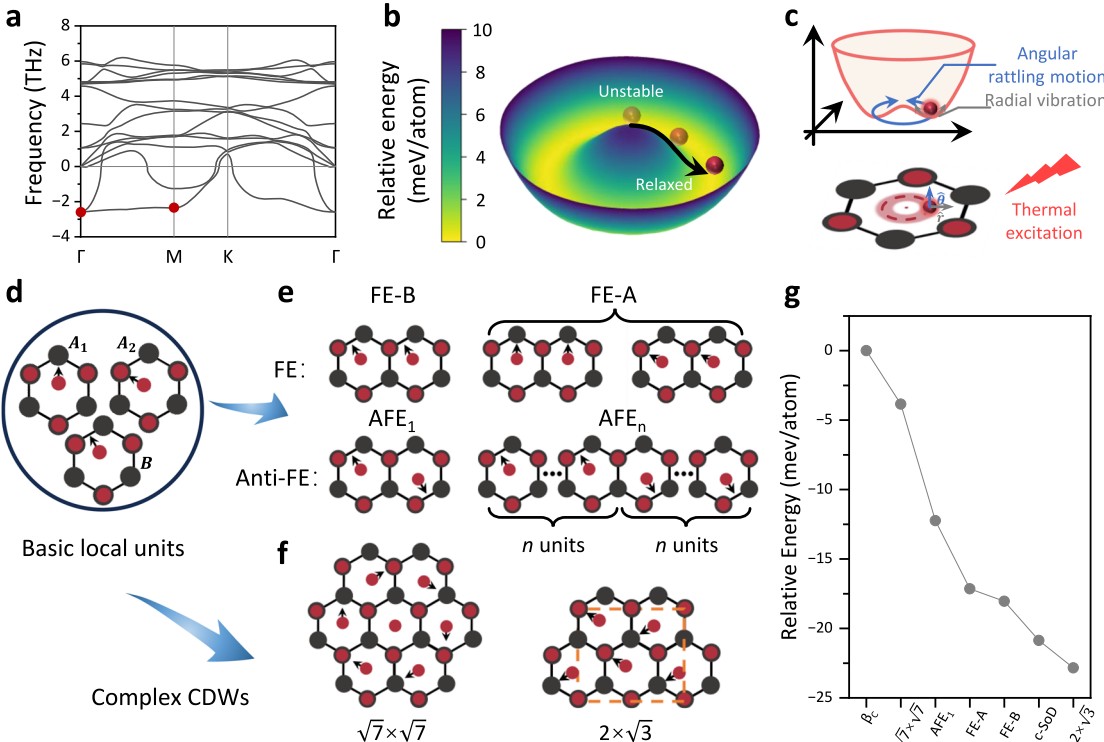

**Fig. 2 | Diversity of CDWs in 2D β-In₂Se₃. a** Phonon band structure of monolayer $\beta_c$-In₂Se₃ with significant imaginary frequencies. Two red dots highlight specific phonon modes which will be discussed later. **b** In-plane Mexican-hat potential energy surface (PES) of Se($m$) atom of 2D $\beta_c$-In₂Se₃[35]. **c** The schematic diagram of the angular rattling motion and radial vibration of Se($m$) atom with a Mexican-hat PES under thermal excitation. **d** Three basic local units of the CDW phase. **e** The ferroelectric (FE) and antiferroelectric (AFE) phases with parallel and antiparallel arrangement of basic local units. **f** The $\sqrt{7} \times \sqrt{7}$ and $2 \times \sqrt{3}$ CDW orders that formed by the specific arrangement of basic local units. **g** The relaxed energies of various CDW phases, using the energy of $\beta_c$ phase as a reference.

by taking the lattice vectors as $\boldsymbol{a}_{CDW} = 4\boldsymbol{a} + \boldsymbol{b}$ and $\boldsymbol{b}_{CDW} = -\boldsymbol{a} + 3\boldsymbol{b}$, where $\boldsymbol{a}$ and $\boldsymbol{b}$ marked by the black arrows in Fig. 1b are the lattice vectors of $\beta_c$ phase, the spatial translational invariance of Se($m$) displacements can be broken, and a new CDW phase can be formed. Figure 1c illustrates the distortions of Se($m$) atoms during the formation of CDW, in which one of the Se($m$) atoms serves as the center of the CDW, denoted as Se($m1$), and its six nearest-neighboring Se($m$) atoms, labeled as Se($m2$) will rotate clockwise (left-handed, LH) or counterclockwise (right-handed, RH), leading to a ferroelectric dipole vortex pattern. At the same time, six Se($m3$) atoms shrink toward the Se($m1$) atom, and finally a $\sqrt{13} \times \sqrt{13}$ c-SoD configuration is formed. Here, the chirality is used to define the rotation direction of Se($m2$) atoms. Figure 1d, e shows the top view of the atomic structure of $\beta_c$ and c-SoD phases. In an actual situation, the LH-c-SoD configuration in Fig. 1e can be formed by the displacement of the atoms enclosed by the gray dashed line in Fig. 1d.

The white and orange arrows in Fig. 1f, g mark the reciprocal lattice vectors of the $\beta_c$ and c-SoD phases of 2D In₂Se₃, respectively, which are extracted by the fast Fourier transform of the corresponding atomic structures. Their relation follows the formula: $\boldsymbol{a}^*_{CDW} = \frac{3}{13}\boldsymbol{a}^* + \frac{1}{13}\boldsymbol{b}^*$, $\boldsymbol{b}^*_{CDW} = -\frac{1}{13}\boldsymbol{a}^* + \frac{4}{13}\boldsymbol{b}^*$, matching the lattice reconstruction in real space. Density functional theory (DFT) simulated scanning tunneling microscope (STM) images of the empty and filled states of $\beta_c$ and c-SoD phases are shown in Fig. 1h–k, the simulated bias voltages are +2 V (Fig. 1h, i) and −2 V (Fig. 1j, k), respectively. It can be seen that the hexagonal lattice and three-fold rotational symmetry of 2D $\beta_c$-In₂Se₃ are still preserved, but the mirror symmetry is broken. Additionally, the lattice constant increases from 4.0 to 14.6 Å after the c-SoD CDW transition. Combining the vertical positions and the partial density of states (PDOS) shown in Supplementary Fig. 2 of Supplementary Note 2, bright spots of the $\beta_c$ phase in Fig. 1h, j are contributed

by the top layer Se [Se($t$)] atoms. However, the Se($t$) atoms of the c-SoD phase will fluctuate in the vertical direction due to CDW distortion, and they can be divided into four categories, named Se($t1$-$4$), as shown in Fig. 1c. The height sequence of these atoms from top to bottom is Se($t2$) > Se($t1$) > Se($t4$) > Se($t3$). Combined with the PDOS in Supplementary Fig. 3 of Supplementary Note 2, the brightness order of the spots in the STM patterns in Fig. 1i, k from bright to dark is Se($t2$) > Se($t1$) > Se($t4$) ≈ Se($t3$). The dynamical stability of the c-SoD phase at 200 K is confirmed through molecular dynamics (MD) simulations in Supplementary Fig. 4 of Supplementary Note 3. In comparison with the SoD-reconstructed CsV₃Sb₅ and 1$T$-TaS₂, whose CDW stable temperatures in experiments are about 94 K and 180 K[36,37], respectively, the 2D β-In₂Se₃ predicted in this study exhibits strong thermal stability.

Then, the question arises: what is the physical origin of this c-SoD? Firstly, considering the high-symmetry $\beta_c$-In₂Se₃ as a semiconductor has a nonzero bandgap, the theory of FSN is not applicable[38]. Furthermore, if we assume that 2D $\beta_c$-In₂Se₃ belongs to the type II CDW, the phonon spectrum is expected to exhibit localized imaginary frequencies at $\boldsymbol{Q}_{CDW}$ due to strong EPC, similar to what is observed in 1$T$-VSe₂ and 2$H$-NbSe₂[39,40]. However, the phonon spectrum analysis in Fig. 2a reveals that 2D $\beta_c$-In₂Se₃ exhibits surprisingly flat and significant imaginary frequencies across a wide range of reciprocal space, with no localized phonon imaginary frequency observed at the $\boldsymbol{Q}_{CDW}$ of the c-SoD phase.

## Diversity of CDWs in 2D β-In₂Se₃

We indicate that although EPC is not the exclusive factor leading to the emergence of c-SoD, the significant flat phonon imaginary frequencies lay a crucial foundation for the multi-CDW orders in 2D β-In₂Se₃. Firstly, DFT calculations indicate that the phonon imaginary frequencies are attributed to the Mexican-hat PES of Se($m$) atoms[35]. As

shown in Fig. 2b, the local maximum of the PES is located at the centrosymmetric position of 2D $\beta_c$-In$_2$Se$_3$. Hence, the Se($m$) atom is expected to deviate from the center site and fall into the basin. Considering the symmetry of the crystal, there should be 12 minima on the basin, and the stable positions of Se($m$) atoms can be categorized into three types as shown in Fig. 2d. According to the distortion direction of Se($m$) atoms, these configurations can be named as $A_1$ (distort towards In atoms), $A_2$ (distort towards Se atoms), and $B$ (distort towards In-Se bonds). Therefore, if each Se($m$) atom within a unit cell slides into the same distorted configuration, ferroelectric (FE) phases with all dipoles consistently aligned can be obtained, as shown by FE-A and FE-B in Fig. 2e. The existence of these FE phases has also been experimentally confirmed[41]. From the perspective of the EPC effect, the formation of FE phases can be attributed to the imaginary phonon modes at the $\Gamma$ point marked by the red dot in Fig. 2a. The eigenmotions of these two degenerate modes involve the in-phase vibrations of Se($m$) atoms in the in-plane direction, as detailed in the Supplementary Fig. 5 of Supplementary Note 4.

However, since the basin of the PES is nearly flat, the Se($m$) atom in each unit cell may distort towards different configurations at finite temperatures. This suggests that c-SoD may not be the only CDW configuration in 2D $\beta$-In$_2$Se$_3$, and the flat imaginary phonon modes may induce the generation of multiple CDW orders. Hence, the translation invariance of the original lattice vectors of 2D $\beta_c$-In$_2$Se$_3$ is broken. Figure 2c illustrates the schematic diagram of Se($m$) atoms independently undergoing angular oscillation along $\hat{\theta}$ and radial vibration along $\hat{r}$ within the basin of the PES under the thermal excitation. Therefore, the spatially uneven fluctuations of each Se($m$) atom can lead to the formation of distinct configurations with local energy minima, subsequently inducing complex CDW orders, such as the aforementioned c-SoD phase. Along these lines, a series of 2D $\beta$-In$_2$Se$_3$ phases can be constructed. The three off-center primitive cells ($A_1$, $A_2$, and $B$) that triggered by the significant imaginary frequencies together with the centrosymmetric cell [i.e., Se($m$) at the center of the Mexican-hat PES, named $C$], can be served as the basic units of the CDW orders. When basic units with opposite distortion directions coexist in equal amounts and orderly manner (such as an equal combination of $B$ and $-B$, where the "minus" denotes the opposite direction of atomic distortion), 2D $\beta$-In$_2$Se$_3$ can exist in the form of an antiferroelectric (AFE) phase, which is also consistent with recent experimental observations[42]. Figure 2e shows the AFE phase with opposite polarizations of adjacent units, and from the perspective of EPC effect, it corresponds to the opposite vibrations of adjacent Se($m$) atoms. For the lowest (imaginary-frequency) phonon branch along the $\Gamma$-M path in Fig. 2a, there are also eigen vibrations of Se($m$) atoms along the lattice vector $a$, but with a phase difference. For the mode marked by the red dot at point M, the neighboring Se($m$) atoms vibrate out of phase, which may correspond to the formation of the AFE phase, the detailed eigen motions are presented in Supplementary Fig. 5 of Supplementary Note 4. In addition to the AFE phase, c-SoD phase with opposite chirality, namely, the antichiral SoD (ac-SoD) phase with similar energies, can also be formed in monolayer $\beta$-In$_2$Se$_3$, the detailed atomic structures are shown in Supplementary Fig. 6 of Supplementary Note 5.

Additionally, through the combination of basic local units with different distortion directions, more complex CDW orders can be formed. For example, we also predict a new $\sqrt{7} \times \sqrt{7}$ CDW phase which is formed by the clockwise/counterclockwise motion of the six nearest neighboring Se($m2$) atoms centered around one Se($m1$) atom, similar to the $\sqrt{13} \times \sqrt{13}$ c-SoD phase. The schematic diagram of the atomic structure is shown in Fig. 2f, it can be seen as a combination of three $A_1$ units, three $A_2$ units, and one $C$ unit. Besides, when the basic units are combined in the form of two $A_1$ units and two $A_2$ units, the in-plane ferroelectric $2 \times \sqrt{3}$ configuration CDW order can be formed. The relaxed atomic structures

of the above CDW orders are presented in Supplementary Fig. 7 of Supplementary Note 6.

For CDW phases with relatively simple configurations, such as the FE phase and AFE phase, we can easily identify soft phonon modes that correspond to their $Q_{CDW}$ in the phonon spectrum (red dots in Fig. 2a). However, for complex CDW orders such as the $\sqrt{13} \times \sqrt{13}$ (c-SoD), $\sqrt{7} \times \sqrt{7}$ and $2 \times \sqrt{3}$, pinpointing the localized unstable phonon modes that correspond to their order vectors becomes challenging, which implies that the complex CDW configuration may be a superposition of multiple soft phonon modes. Finally, we compared the energies of different phases in Fig. 2g, all of which exhibit significantly lower energies than the undistorted $\beta_c$ phase. It is worth noting that the AFE phases are stripe-like, and their total energies also depend on the width of stripe[43]. In Fig. 2g, we show the energy of the AFE$_1$ phase as a representative. A more comprehensive energy comparison of the AFE phases is provided in Supplementary Fig. 8 of Supplementary Note 6. In addition, the energies of the $2 \times \sqrt{3}$ phase and the c-SoD phase are very close to each other. Considering that the $2 \times \sqrt{3}$ CDW phase has been experimentally confirmed to be stable at 170 K[44]. Therefore, we predict that the c-SoD phase may also be experimentally stable in a similar temperature range, which is also consistent with the stability deduced from the MD in Supplementary Fig. 4. The current experimental observations of the in-plane polarized $2 \times \sqrt{3}$ phase are achieved by cooling the FE $\beta$ phase, which also has an in-plane ferroelectricity as the seed for the transition. However, to obtain the centrosymmetric c-SoD phase (with zero net electric dipole moment) in experiments, we suggest one should eliminate the ferroelectric seed in the FE $\beta$ phase before cooling the temperature, possibly with effective methods like optically electronic excitation or electron doping[45].

## Origin of CDW in 2D $\beta$-In$_2$Se$_3$

In addition to the periodic modulation of the charge density, the significant opening of the bandgap is another important feature of CDWs in 2D $\beta$-In$_2$Se$_3$. Here, taking the c-SoD phase of 2D In$_2$Se$_3$ as an example, we calculated its PBE band structure before and after the CDW transition, respectively. For a better comparison, both calculations were conducted using a $\sqrt{13} \times \sqrt{13}$ reconstructed cell. The results in Fig. 3a, b show that after the c-SoD CDW distortion, the PBE energy bandgap increases from 0.49 eV to 1.38 eV. Unlike previously reported SoD CDW materials, such as 1$T$-NbSe$_2$ whose bandgap opening is induced by the splitting of $d$-orbitals-related Hubbard band[14], the electronic structure of 2D $\beta$-In$_2$Se$_3$ is relatively simple. The PDOS in Fig. 3a, b show that the orbitals near the Fermi level consist only of In-$s,p$, and Se-$p$ states. To further demonstrate that the $d$-orbitals are not necessary for the formation of CDW, we performed structural relaxation and electronic band structure calculations of the c-SoD phase but used the pseudopotential that excludes the 4$d$-orbitals of In atoms. As shown in Supplementary Fig. 9 of Supplementary Note 7, the band structure is consistent with that (considering the 4$d$-orbitals of In atoms) in Fig. 3b.

Furthermore, we conducted integrated crystal orbital bond index (ICOBI) analysis and charge density difference (CDD) calculations, attempting to provide insights into the bandgap increase from the perspectives of changes in electron localization and local covalency of In-Se bonds. ICOBI is an intuitive parameter to quantify the covalency of chemical bonds, with the value closer to 1 indicating stronger covalency[46]. Figure 3c shows the ICOBI of In-Se bonds in the c-SoD phase, with gray scatter points indicating the In($t$ or $b$)-Se($t$ or $b$) bonds and red scatter points representing the In($t$ or $b$)-Se($m$) bonds. The In($t$ or $b$)-Se($t$ or $b$) and In($t$ or $b$)-Se($m$) bonds of pristine 2D $\beta_c$-In$_2$Se$_3$ used for reference are marked with gray and red pentagrams, respectively. After the c-SoD CDW transition, the bond length of In($t$ or $b$)-Se($t$ or $b$) slightly changes from 2.68 Å to a range of 2.62–2.79 Å, and the ICOBI changes from 0.69 to a range of 0.56–0.83. In contrast, the In($t$ or $b$)-

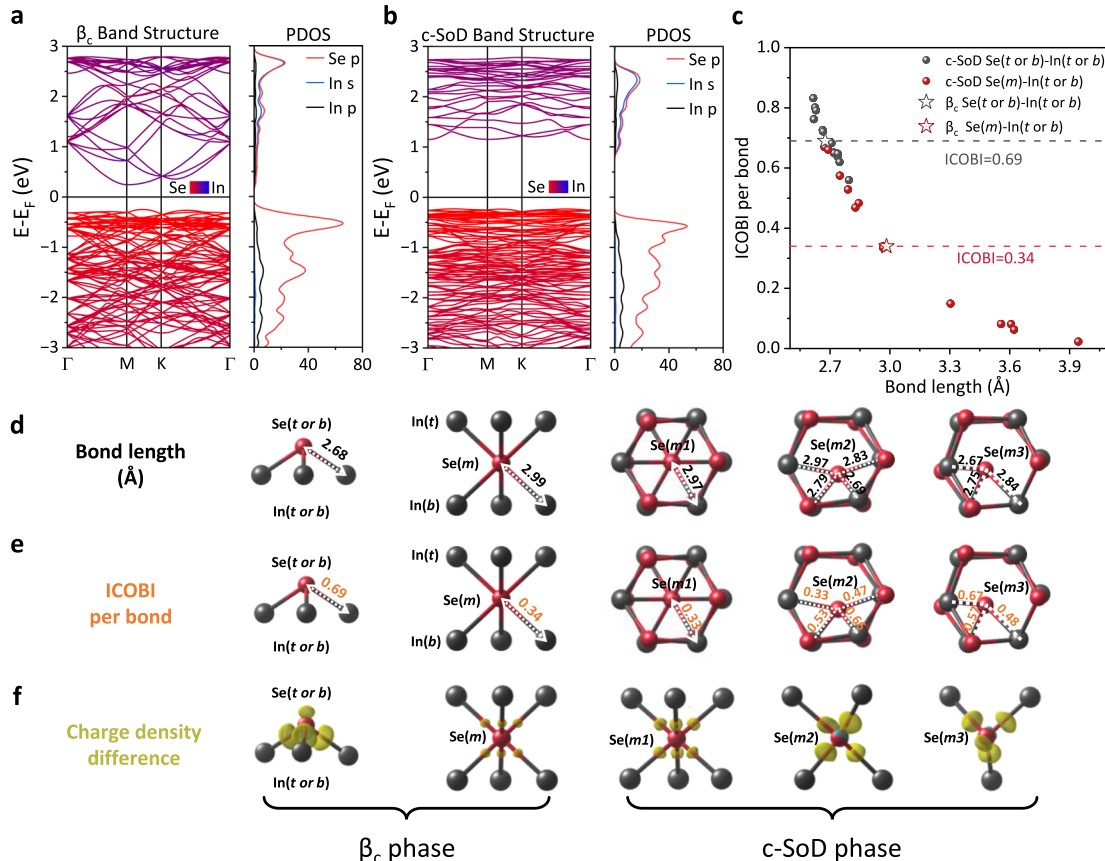

**Fig. 3 | Changes in electronic structure and chemical bonds of 2D β-In$_2$Se$_3$ before and after the c-SoD CDW transition.** PBE band structures and partial density of states (PDOS) of the (**a**) β$_c$ phase (using the $\sqrt{13} \times \sqrt{13}$ supercell) and the (**b**) $\sqrt{13} \times \sqrt{13}$ c-SoD CDW phase of 2D In$_2$Se$_3$. **c** Integrated crystal orbital bond index (ICOBI) (per bond) varies with the length of In-Se bond in the c-SoD phase, the red and gray dashed lines mark the ICOBI of the In(*t* or *b*)-Se(*m*) and In(*t* or *b*)-Se(*t* or *b*) bonds in the β$_c$ phase, respectively. **d–f** Bond length, ICOBI, and the local charge density difference (CDD) of the β$_c$ and c-SoD CDW phases. The CDD is defined as $\Delta\rho = \rho_{scf} - \rho_{atom}$, where $\rho_{scf}$ represents the charge density of In$_2$Se$_3$ compound and $\rho_{atom}$ is the superposition of charge density of isolated In and Se atoms. The displaying In-Se bond length cutoff is 3 Å. The isosurface value of CDD is 0.0065 $e/a_0^3$, $a_0$ is the Bohr radius. Yellow isosurface represents charge accumulation.

Se(*m*) bond has undergone more significant changes, with length changing from 2.99 Å to the range of 2.67–3.94 Å, and the ICOBI changing from 0.34 to the range of 0.02–0.67. This illustrates the dominant role of Se(*m*) in the formation of CDW, highlighting the importance of the Mexican-hat PES and the flat soft phonon modes. Figure 3d–f mark the length, ICOBI, and CDD of In-Se bonds in the β$_c$ and c-SoD phases with a 3.00 Å displaying bond length cutoff. These results indicate that the CDW transition is accompanied by the break of a significant number of In(*t* or *b*)-Se(*m2* or *m3*) bonds. For those In(*t* or *b*)-Se(*m*) bonds whose lengths are shortened, the concentration of charge on the bonds increases, leading to an enhancement in electron localization. Simultaneously, the ICOBI of these In(*t* or *b*)-Se(*m*) bonds also increases, indicating strengthened covalency. The complete CDD images of β$_c$ and c-SoD phase are shown in Supplementary Fig. 10 of Supplementary Note 8.

Based on the above analysis, it can be concluded that the formation of CDW is inseparable from the distortion of Se(*m*) atoms and the enhancement of electron localization (or local covalency). On one hand, the distortions of Se(*m*) atoms are empowered by their intrinsic Mexican-hat PES and flat soft phonon modes, which introduce a new degree for regulating the configuration of CDW. On the other hand, the electron localization and local covalency of chemical bonds in the specific configuration is significantly enhanced, resulting in bandgap opening and a decrease in energy, thereby locking the CDW order and maintaining its stability. Furthermore, we calculated the PBE band structures of the other six CDW configurations

mentioned in Fig. 2e, f. The results in Supplementary Fig. 11 of Supplementary Note 9 exhibit a significant increase in bandgap after the CDW transitions.

## Universality of CDW in 2D β-III$_2$−VI$_3$

Considering that the Mexican-hat PES was confirmed to be a general phenomenon in 2D β$_c$-III$_2$−VI$_3$ materials (III = Al, Ga, In; and VI = S, Se, Te)[35], then CDW in 2D In$_2$Se$_3$ should also be prevalent in the other eight materials. Through structural relaxation, we indeed identified the other eight c-SoD CDW configurations. The local atomic structures and the DFT-simulated STM patterns are shown in Fig. 4a–h, the detailed atomic structures are presented in Supplementary Fig. 12 of Supplementary Note 10. The STM patterns of these eight materials all exhibit the same $\sqrt{13} \times \sqrt{13}$ lattice periodicity and three-fold symmetry as those of the c-SoD phase of 2D In$_2$Se$_3$. Previous studies have suggested that CDW transitions are often accompanied by a metal-to-insulator transition. However, the situation is more diverse in 2D III$_2$−VI$_3$ compounds. Here, using the higher-level hybrid functional (HSE06), we calculated the band structures of nine 2D III$_2$−VI$_3$ compounds. The bar charts in Fig. 4i indicate that the bandgaps of their β$_c$ phase range from 0 to 2.4 eV, while the bandgaps of the nine c-SoD CDW phases are in the range of 0.9−3.5 eV, suggesting that there are both metal-to-insulator and insulator-to-insulator CDW transitions in 2D III$_2$−VI$_3$. The PBE and HSE06 band structures of the β$_c$ and c-SoD phases of 2D III$_2$−VI$_3$ materials are also shown in Supplementary Figs. 13-16 of Supplementary Note 11.

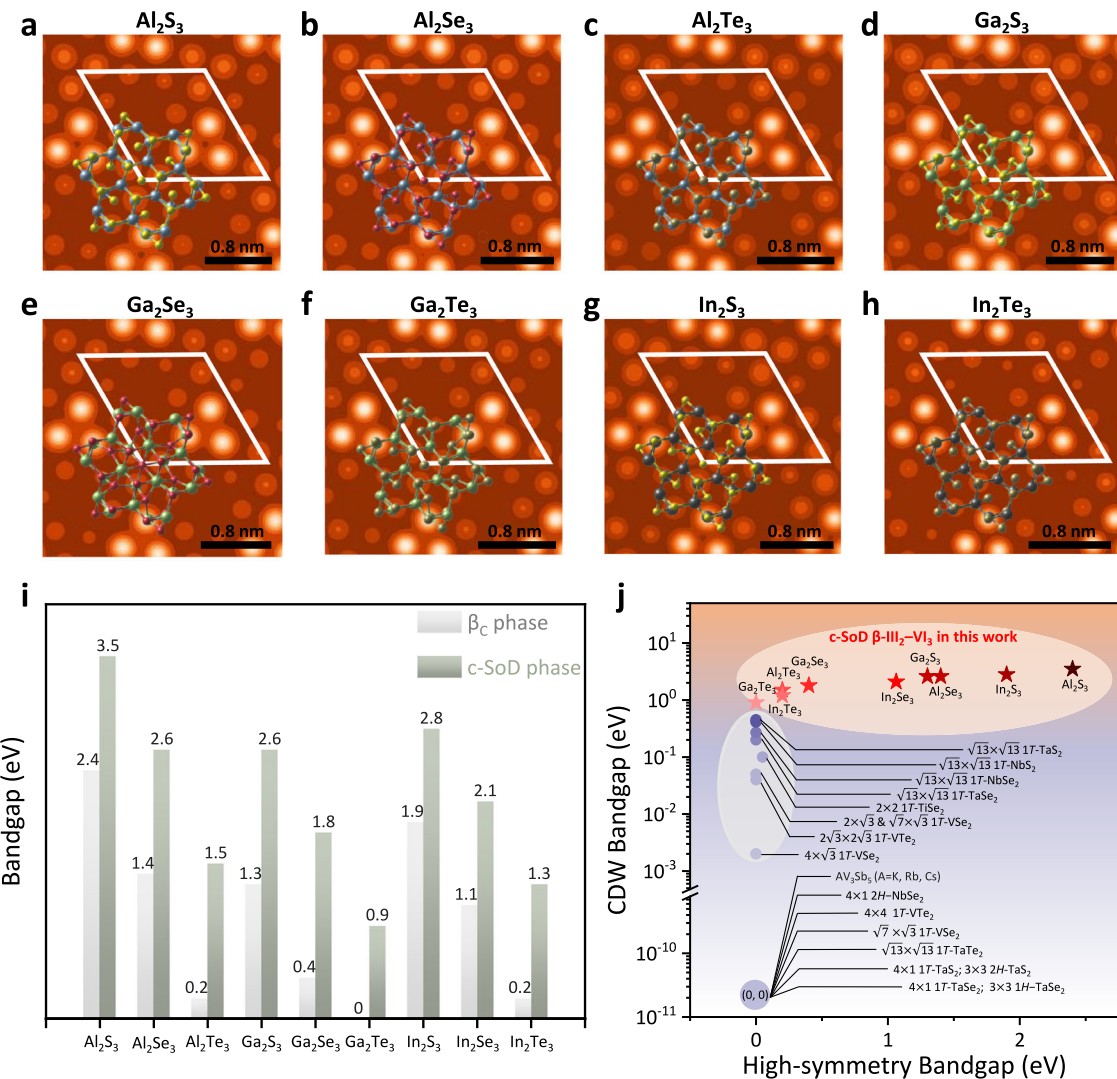

**Fig. 4 | Electronic-property map of the 2D III₂–VI₃ CDWs and the comparison with other reported 2D CDW materials. a–h** Simulated STM patterns of the $\sqrt{13} \times \sqrt{13}$ c-SoD CDW phase in 2D III₂–VI₃ family. **i** The HSE bandgaps of high-symmetry $\beta_c$ phase (gray) and the c-SoD CDW phase (green) of 2D III₂–VI₃. **j** Electronic-property map of 2D III₂–VI₃ CDW and previously reported 2D or quasi− −2D CDW materials, the horizontal axis indicates the bandgap of the pristine high-symmetry phase, and the vertical axis indicates the bandgap of the CDW phase. The purple circle marked with coordinates (0, 0) encompasses materials in which the bandgap remains zero both before and after the CDW transition. All the bandgaps of the previously reported materials are obtained from the published literature[14,47–59].

Compared to traditional 2D CDW materials such as transition metal dichalcogenides (TMDs) that are represented by VTe₂ and NbSe₂, the most distinctive feature of 2D III₂–VI₃ materials is a more significant gap opening during the CDW transitions (by ranging from 0.9 to 1.4 eV), which further supports the inference that electron localization enhancement plays an important role in the formation of CDW. In Fig. 4j, a detailed representation of the electronic-property map for previously reported 2D and quasi−2D CDW materials is provided. The purple circle located at the bottom of the map marked with coordinates (0, 0) encompasses materials in which the bandgap remains zero both before and after the CDW transition. Although, usually a $k$-dependent non-zero CDW gap can be defined in these systems[47–54]. The purple dots on the gray oval background represent the material in which the bandgap really opens after the CDW transition, but the bandgap opening is quite small and in a range of 0.04–0.6 eV[14,53–59]. We also marked 2D III₂–VI₃ materials with red pentagon stars on the map, whose CDW transitions mainly appear as insulator-to-insulator transitions, except in the case of 2D Ga₂Te₃. The large bandgaps of 2D III₂–VI₃ materials position them independently at the top of the CDW electronic-property map, which expands the territory of the 2D CDW family.

## Discussion

Through first-principles calculations, STM simulations, and MD simulations, the $\sqrt{13} \times \sqrt{13}$ c-SoD reconfigured CDW phase is first discovered in 2D β-In₂Se₃ and the entire family of 2D β-III₂–VI₃. Different from traditional SoD CDW materials such as NbSe₂ or TaS₂, which have electrons in $d$-orbitals that exhibit strong correlation characteristics, the electronic structures of 2D III₂–VI₃ materials are rather simple with only $s$- and $p$-orbitals observed near the Fermi-level. However, due to the flat and extensive imaginary phonon frequencies and the Mexican-hat PES of the Se($m$) atoms, the CDW orders of 2D β-In₂Se₃ are complex and diverse, which bring physical phenomena such as atomic-scale ferroelectric vortex in the c-SoD CDW. Through the analysis of the electronic structure and chemical bonds, we found that specific CDW configurations can be stabilized by the localized enhancement of covalency induced by the distorted Se($m$) atoms. Simultaneously, the CDW transition is accompanied by an increased degree of electron localization, which results in significantly wider bandgaps in 2D III₂–VI₃ materials compared to traditional 2D CDW materials. Our work provides a new branch of CDW materials and expands its electronic-property map, which may bring new insights and platforms for future

study on the origin of CDWs, and even the design of next-gen electronic devices.

## Methods

### DFT calculations

In this study, the Vienna ab initio simulation package (VASP) was used for DFT calculations[60]. The projector augmented wave (PAW) pseudopotential and the Perdew–Burke–Ernzerhof (PBE) exchange-correlation functional were employed[61,62]. The $4d$, $5s$, and $5p$ orbitals of In atoms, and the $4s$ and $4p$ orbitals of Se atoms are considered. The Monkhorst-Pack k-points grid for geometry optimization of the high-symmetry $\beta_c$ and the $\sqrt{13} \times \sqrt{13}$ c-SoD phases of 2D III$_2$–VI$_3$ materials are $13 \times 13 \times 1$ and $5 \times 5 \times 1$, respectively. The k-points grids for static energy calculation of the two phases are $17 \times 17 \times 1$ and $6 \times 6 \times 1$, respectively. It is noteworthy that the crystal cell of the high-symmetry 2D $\beta_c$-In$_2$Se$_3$ was also enlarged to a $\sqrt{13} \times \sqrt{13}$ supercell for a better comparison when performing PBE band structure calculations. The energy cutoff for geometry optimization and static calculations was 311 eV. To obtain the more accurate bandgaps of 2D β-III$_2$–VI$_3$ materials before and after the CDW distortions, the Heyd-Scuseria-Ernzerhof hybrid functional (HSE06)[63] with a mixing parameter of 25% was used. A vacuum region with thickness of 20 Å is constructed in the vertical direction of all cells to reduce the influence of periodic boundaries, and the $c$-axis of the cell was fixed during the geometry optimization calculations. A $2 \times 2 \times 1$ supercell of the $\sqrt{13} \times \sqrt{13}$ c-SoD phase containing 260 atoms was constructed for MD simulations and the NVT ensemble was employed[64]. The timestep was set to 1 fs and the energy cutoff was 250 eV. The ICOBI of In-Se bond is calculated using the LOBSTER package[65]. The simulated STM patterns were obtained using the VASP package and the VASPKIT[66] processing program. The phonon band structure is calculated using the Phonopy code[67], and the convergence tests are given in Supplementary Fig. 1. Further details about the calculations are provided in Supplementary Note 1.

## Data availability

The data supporting the findings of this work are available within the article and Supplementary Information. All data are available from the corresponding authors.

## Code availability

The DFT calculations have been carried out using the VASP software[60]. The simulated STM patterns were obtained using the VASPKIT code[66]. The ICOBI of In-Se bond is calculated using the LOBSTER package[65].

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

## Acknowledgements

This work was supported by the National Science and Technology Major Project of China (Grant No. 2022ZD0117600, X.-B.L.), the National Natural Science Foundation of China (Grants No. 12274172, X.-B.L. and 12274180, N.-K.C.), the China Postdoctoral Science Foundation (Grant No. BX20240136, Y.-T.H. and GZB20240341, Z.-Z.L.), and the Natural Science Foundation of Jilin Province (20230101007JC, N.-K.C.). High-Performance Computing Center (HPCC) at Jilin University is also acknowledged. The authors thank Chen Si for helpful discussions. Z.-Z. L. is grateful for the support by the Shuimu Tsinghua Scholar Program.

## Author contributions

Y.-T.H. and Z.-Z.L. predicted the CDW of 2D $III_2$–$VI_3$. Y.-T.H., X.-B.L., N.-K.C., and Z.-Z.L conceived the calculations, Y.-T.H. performed the calculations. Y.-T.H., Z.-Z.L., X.-B.L., N.-K.C., and S.Z. did the theoretical analysis. Y.-T.H., Z.-Z.L., and X.-B.L. wrote the initial draft. H.-B.S. and Y.W. discussed the results and reviewed the manuscript. X.-B.L. and N.-K.C. supervised the project.

## Competing interests

The authors declare no competing interests.
