## [Peer Review File · Nature Communications]

REVIEWER COMMENTS

Reviewer #1 (Remarks to the Author):

Huang et al. have performed a careful and thorough density functional theory based calculation to illustrate a new charge density wave mechanism. They track the flat imaginary phonon mode for the β -In₂Se₃ (and other similar materials), and predict an interesting chiral CDW mode. This results in both left-handed and right-handed enantiomers. Their PBE based band dispersion calculations also suggest a wider bandgap in the chiral CDW phase, as compared with the intact structure. The multiple CDW (FE, AFE, and chiral) modes are subject to the flat imaginary mode, which gives a Goldstone-like Mexican hat energy landscape in the configurational space. This is a promising result, and I believe that both $\sqrt{7}\times\sqrt{7}$ and $\sqrt{13}\times\sqrt{13}$ phases are potentially observed. Before recommending its publication in Nature Communications, I have the following minor suggestions that may further improve the work.

1) The total energy comparison (Figure 2g) might be incomplete. The AFE systems (β -In₂Se₃) are stripe-like, and their total energy depends on the stripe width (e.g., JPCL 2023, 14, 677). These structures have been shown in the Supplementary Information, but their total energy should be compared. Note that the α phase is always more stable than the β phase, while the latter one is metastable and can be experimentally realized.

2) What specific symmetry components are broken during the phase transition from the high symmetric parental β phase to the chiral phase. It seems that the C_{3z} is preserved. If not, then there might be more chiral CDW configurations?

3) The Star-of-David structure here is different from that of the TaS₂, which is non-chiral. Hence, is it possible to form "chiral domains" when two SoDs with opposite chirality meet together? Do they favor in energy or not?

In all, I believe that this new chiral CDW phase is very promising and would exhibit novel physical responses (with symmetry reduction).

Reviewer #2 (Remarks to the Author):

The manuscript authored by Huang et al. presents a systematic study on the spontaneous lattice deformation phenomena known as Charge Density Waves (CDWs), with a focus on a novel family of two-dimensional III₂-VI₃ van der Waals materials. The key result of this research is the revelation of multiple CDW orders in these materials, embodied by the chiral Star of David (SoD) configuration. This finding is interesting, considering that the involved electronic systems are relatively straightforward, comprising solely III-s and VI-p orbitals. Despite their simplicity, the CDWs exhibit

complex transitions between metal-to-insulator and insulator-to-insulator states, with a notable increase in the bandgap. This behavior opposes traditional narrations limited to Fermi surface nesting or strong electron-phonon coupling. The study utilizes first-principles calculations to demonstrate a substantial presence of flat imaginary frequencies across the optical phonon branch, promoting diverse CDW phases. Additionally, it displays a rare phenomenon in the 2D materials, where the CDW vector does not always correlate to local minima in the phonon branch, suggesting an unconventional pathway of CDW formation that extends beyond standard electron-phonon coupling constraints. Overall, this manuscript will be helpful for researchers particularly in the exploration of CDWs in 2D materials. However, I am somewhat reserved in my recommendation to publish this article until the authors address the following concerns.

a) The theoretical findings in this study are generally consistent. However, it would be better to have an experimental verification to support the findings. Is it possible for the authors to conduct a proof-of-concept experiment?

b) On the theoretical part, van der Waals interaction is essential for 2D III2-VI3 van der Waals materials. However, the DFT calculations did not include van der Waals interaction. The authors need to conduct a proper DFT calculation with exchange correlated functionals including van der Waals interaction, such as +D3, +rVV10, +TS, etc.

c) For the phonon calculation, the reviewer did not see any computational details, which is not acceptable. Also, the vdW should be concerned for phonon calculations. A thorough convergence test should be conducted for phonons as it is sensitive to computational parameters.

d) The substantial imaginary frequency of phonon band structure should be analyzed. The detailed eigen motions at the two negative modes at M and the degenerate one at G should be presented.

f) The DFT calculations are not well documented. This is essential for others to reproduce your results. This includes the details of MD simulations and electronic structure calculations. Was the HSE06 or HSE03 used for band structure calculations? What is the mixing parameters? Why did the authors choose HSE over metaGGA (e.g. TB-mBJ) for band gap evaluation?

Reviewer #3 (Remarks to the Author):

In this paper, the authors utilize first-principles calculations, STM simulations, and MD simulations to identify a new charge density wave (CDW) phase in 2D beta-In₂Se₃. They suggest that the CDW does not stem from strong electron-electron interactions but is instead due to the flat and extensive imaginary phonon frequencies and the Mexican-hat potential energy surface (PES) of the Se(m) atoms. If validated, these findings would be significant, offering fresh insights and new avenues for investigating the origins of CDWs. The manuscript is well-written overall. However, a critical shortcoming is the lack of experimental evidence for the new CDW phase, specifically the chiral star of David (c-SOD) in 2D In₂Se₃ (and other 2D III₂-VI₃ materials). The paper also fails to provide a plausible explanation for the absence of experimental observations. Additionally, there are several unclear aspects in the manuscript, as outlined below. If the authors can adequately address these issues, I would recommend publication in Nature Communications.

1. What new physical properties are associated with the c-SOD phase?
2. The authors selected In₂Se₃, which is known for its complex phase transition behavior. What are the main differences among 2D III₂-VI₃ materials? Are similar CDW phases, such as the c-SOD phases, found in other 2D III₂-VI₃ materials?
3. Given the slight relaxed energy difference between the $\sqrt{13} \times \sqrt{13}$ and $2\sqrt{3}$ phases (see Fig. 2g), why has the $\sqrt{13} \times \sqrt{13}$ phase not been observed experimentally, while the $2\sqrt{3}$ phase has?
4. In Fig. 1c, the labels for left-hand and right-hand are difficult to distinguish. I suggest using different colors and/or line types to make them more distinguishable.

Point-by-Point Response to Reviewers' Comments

Reviewer #1 (Remarks to the Author):

General Comment: Huang et al. have performed a careful and thorough density functional theory based calculation to illustrate a new charge density wave mechanism. They track the flat imaginary phonon mode for the β - In_2Se_3 (and other similar materials), and predict an interesting chiral CDW mode. This results in both left-handed and right-handed enantiomers. Their PBE based band dispersion calculations also suggest a wider bandgap in the chiral CDW phase, as compared with the intact structure. The multiple CDW (FE, AFE, and chiral) modes are subject to the flat imaginary mode, which gives a Goldstone-like Mexican hat energy landscape in the configurational space. This is a promising result, and I believe that both $\sqrt{7}\times\sqrt{7}$ and $\sqrt{13}\times\sqrt{13}$ phases are potentially observed. Before recommending its publication in Nature Communications, I have the following minor suggestions that may further improve the work.

Reply to General Comment: We sincerely thank the referee very much for the positive evaluation of our work. The detailed responses to your comments are shown below.

Comment 1: The total energy comparison (Figure 2g) might be incomplete. The AFE systems (β - In_2Se_3) are stripe-like, and their total energy depends on the stripe width (e.g., JPCL 2023, 14, 677). These structures have been shown in the Supplementary Information, but their total energy should be compared. Note that the α phase is always more stable than the β phase, while the latter one is metastable and can be experimentally realized.

Reply to Comment 1:

Figure R1. (a) Atomic structures of the anti-ferroelectric (AFE) phase of monolayer β - In_2Se_3 with different domain widths, namely AFE_n ($n=1, 2, 3, 4$). The dash lines highlight the unit cells for these AFE_n phases. (b) The relative energy of AFE_n phases with different domain widths.

Thanks to the reviewer for providing reference, which give us more inspiration. According to the suggestion, we compare the relative energy of the antiferroelectric (AFE) phases with different stripe widths in Figure R1. These phases are named as AFE_n , $n=1, 2, 3, 4$, respectively, according to their stripe width. The calculated results show that the AFE_1 phase has the highest energy and the

AFE₃ phase is the most energetically favorable phase, which is consistent with the conclusions given in the reference [J. Phys. Chem. Lett. 14, 677-684 (2023)]. According to the reference, the difference in energy can be attributed to the different concentrations of ferroelectric domain walls. For the AFE₁ phase, a higher concentration of domain wall induces a more significant structural distortion. When the width of the stripe increases, the concentration of the domain wall decreases and the total energy tends to stabilize and be comparable to that of the c-SoD and the $2\times\sqrt{3}$ phases.

In response to your comment, we have added **Figure R1** to **Supplementary Information** and added a discussion in the main text. Also, we thank the reviewer for kindly reminding us that the α phase is more stable. According to our previous calculations, the energy of the α phase is indeed always lower than that of the β family at 0 K. However, when the temperature is raised (such as, to 750 K), the α phase will transform into the pseudo-centrosymmetric β phase, which has a significant configuration entropy [Appl. Phys. Rev. 8, 031413 (2021)]. Hindered by the entropy barrier of the β -to- α transition, the pseudo-centrosymmetric β phase remains stable over a wide temperature range and can be observed experimentally.

Detail changes in the main text (highlighted in red) include:
Page 10-11

It is worth noting that the AFE phases are stripe-like, and their total energies also depend on the width of stripe [J. Phys. Chem. Lett. 14, 677-684 (2023)]. In Fig. 2g, we show the energy of the AFE₁ phase as a representative. A more comprehensive energy comparison of the AFE phases is provided in Supplementary Fig. 8 of Supplementary Note 6.

Comment 2: What specific symmetry components are broken during the phase transition from the high symmetric parental β phase to the chiral phase. It seems that the $C3_z$ is preserved. If not, then there might be more chiral CDW configurations?

Reply to Comment 2:

Figure R2. Schematic diagram of the symmetry of monolayer In₂Se₃ before and after the c-SoD CDW transition.

Thank you to the reviewer for the insightful comment. The pristine monolayer $\beta\text{-In}_2\text{Se}_3$ (β_c phase) belongs to the achiral space group $P\bar{3}m1$ (No. 164) and exhibits three-fold rotational symmetry (C_{3z}), mirror symmetry, and inversion symmetry, as shown by the schematic in **Figure R2(a)**. The emergence of c-SoD configuration requires the clockwise or counterclockwise rotation of specific $\text{Se}(m)$ atoms, which breaks the mirror symmetry of the parental β_c phase while preserving three-fold rotational symmetry (C_{3z}) and inversion symmetry, finally induces the in-plane chirality. Therefore, the chiral structures resulting from the breaking of C_{3z} symmetry are not observed in our study, and a deeper investigation into the relationship between chiral structure, symmetry breaking, and CDW orders will be conducted in our future research.

Comment 3: The Star-of-David structure here is different from that of the TaS_2 , which is non-chiral. Hence, is it possible to form "chiral domains" when two SoDs with opposite chirality meet together? Do they favor in energy or not?

Reply to Comment 3:

Figure R3. Relaxed atomic structures of chiral domains of LH and RH c-SoD phases with different stripe widths, namely ac-SoD_n ($n=1, 2, 3$).

We thank the reviewer for the illuminating question, based on which we performed the following calculations. Similar to the ferroelectric phase, the chiral SoD (c-SoD) phase of In_2Se_3 also exhibits directional properties due to the left-handed (LH) rotation and right-handed (RH) rotation modes of the middle layer Se atoms. Therefore, similar to the analyses in Reply to Comment 1 regarding the AFE_n phases, we constructed supercells that simultaneously encompass both LH c-SoD and RH c-SoD with different stripe widths, i.e., antichiral SoD [namely, ac-SoD_n, $n=1-3$]. The geometrically optimized atomic structures are shown in **Figure R3**, all these configurations can be sustained,

indicating their potential stability.

The energies of the three ac-SoD_n (n=1-3) phases, along with the single chirality c-SoD phase, are presented in **Table R1**. Unlike the AFE phases discussed in Reply to Comment 1, whose energies vary with the width of the AFE stripe, the energies of three ac-SoD_n phases are comparable and similar to that of the single chirality c-SoD phase. This phenomenon can be attributed to the fact that, when two domains with opposite chirality meet together, the boundary is primarily composed of Se(*m3*) and their surrounding atoms. However, the Se(*m3*) atoms are all contracted towards Se(*m1*) atoms in a chirality-independent manner, which prevents structural distortions similar to those observed in the domain wall of AFE phases. Therefore, the presence of chiral domains has no significant effect on the total energy. Inspired by this comment, we added a discussion of the ac-SoD phase to the main text and added **Figure R3** and **Table R1** to **Note 5** of the **Supplementary Information**.

Table R1. Total energies (per atom) of the c-SoD and ac-SoD phases with different stripe widths.

Phase	c-SoD	ac-SoD ₁	ac-SoD ₂	ac-SoD ₃
Total energy (eV/atom)	-3.687	-3.688	-3.688	-3.688

Detail changes in the main text (highlighted in red) include:

Page 10

In addition to the AFE phase, c-SoD phase with opposite chirality, namely, the antichiral SoD (ac-SoD) phase with similar energies, can also be formed in monolayer β -In₂Se₃, the detailed atomic structures are shown in Supplementary Note 5.

Comment 4: In all, I believe that this new chiral CDW phase is very promising and would exhibit novel physical responses (with symmetry reduction).

Reply to Comment 4: We thank the referee very much once again for the very high evaluation of our work.

Reviewer #2 (Remarks to the Author):

General Comment: The manuscript authored by Huang et al. presents a systematic study on the spontaneous lattice deformation phenomena known as Charge Density Waves (CDWs), with a focus on a novel family of two-dimensional $\text{III}_2\text{-VI}_3$ van der Waals materials. The key result of this research is the revelation of multiple CDW orders in these materials, embodied by the chiral Star of David (SoD) configuration. This finding is interesting, considering that the involved electronic systems are relatively straightforward, comprising solely III-s and VI-p orbitals. Despite their simplicity, the CDWs exhibit complex transitions between metal-to-insulator and insulator-to-insulator states, with a notable increase in the bandgap. This behavior opposes traditional narrations limited to Fermi surface nesting or strong electron-phonon coupling. The study utilizes first-principles calculations to demonstrate a substantial presence of flat imaginary frequencies across the optical phonon branch, promoting diverse CDW phases. Additionally, it displays a rare phenomenon in the 2D materials, where the CDW vector does not always correlate to local minima in the phonon branch, suggesting an unconventional pathway of CDW formation that extends beyond standard electron-phonon coupling constraints. Overall, this manuscript will be helpful for researchers particularly in the exploration of CDWs in 2D materials. However, I am somewhat reserved in my recommendation to publish this article until the authors address the following concerns.

Reply to General Comment: We appreciate the referee's positive evaluation of our work. Here, we address the comments from the referee one by one, as below.

Comment 1: The theoretical findings in this study are generally consistent. However, it would be better to have an experimental verification to support the findings. Is it possible for the authors to conduct a proof-of-concept experiment?

Reply to Comment 1: We would like to thank the reviewer for affirming the findings of our theoretical work. Due to the limitations of experimental conditions, indeed we are still unable to complete experimental verification in the short term. However, in the revised manuscript we have outlined a theoretical strategy to assist in the experimental observation of the predicted c-SoD phase, please also see **the Reply to comment 3 of Reviewer #3**.

Comment 2: On the theoretical part, van der Waals interaction is essential for 2D $\text{III}_2\text{-VI}_3$ van der Waals materials. However, the DFT calculations did not include van der Waals interaction. The authors need to conduct a proper DFT calculation with exchange correlated functionals including van der Waals interaction, such as +D3, +rVV10, +TS, etc.

Reply to Comment 2: Thank the reviewer for the advice. Although $\text{III}_2\text{-VI}_3$ materials are van der Waals (vdW) layered materials, the calculations in our study are entirely focused on the monolayer structures. When constructing the atomic models, we also included a vacuum layer with a thickness of 20 Å along the out-of-plane direction in our calculated cell. Therefore, compared to real multilayer materials, the vdW interaction in our work (monolayer case) should be very weak. To

ensure the rigor of the predicted c-SoD CDW, the geometry optimization calculation that considered D3 vdW interaction was also conducted for the monolayer c-SoD In_2Se_3 . The optimized structures without and with D3 vdW interaction corrections are shown in **Figure R4**. The optimized structures under both conditions exhibit the same c-SoD configuration, demonstrating the robustness of the c-SoD CDW in monolayer In_2Se_3 .

Figure R4. The optimized atomic structure of monolayer In_2Se_3 (a) without and (b) with the D3 vdW interaction corrections, both exhibiting the same c-SoD configuration that marked by the orange hexagams.

Figure R5. (a) Illustrations of the two stacking sequences (AA and AB stacking) of bilayer c-SoD In_2Se_3 . (b) The static total energy as a function of interlayer spacing under two stacking sequences.

Inspired by the comment, we also performed DFT geometry optimization of bilayer c-SoD In_2Se_3 , in which the D3 vdW interaction corrections are necessary. As shown in **Figure R5(a)**, two kinds of staking configurations are considered: the first configuration, where the $\text{Se}(m1)$ atom in the upper quintuple layer aligns with one $\text{Se}(t1)$ atom of the lower quintuple layer, resembles the AB-stacking in bulk $\beta\text{-In}_2\text{Se}_3$ [Phys. Rev. B 98, 16, 165134 (2018)]; for AA-staking, the $\text{Se}(m1)$ atoms in the upper and lower quintuple layers align with each other. We compared the energies of bilayer c-SoD In_2Se_3 across a wide range of interlayer spacings in the forms of the two stacking sequences. As shown in **Figure R5(b)**, the energy of AB-stacking is significantly lower than that of AA-stacking, indicating that AB stacking is more stable. The geometric optimization of the AB-stacking bilayer In_2Se_3 that considers the D3 vdW interaction corrections was conducted. The relaxed atomic structure is shown in **Figure R6**. It can be seen that both the upper and lower quintuple layers clearly

exhibit the c-SoD configuration, demonstrating that the c-SoD CDW is still robust in the bilayer β - In_2Se_3 where the D3 vdW interactions are considered.

Figure R6. Atomic structure of the AB-stacking bilayer c-SoD In_2Se_3 , with considering the D3 vdW interaction. The top views of the upper and lower quintuple layers are shown on the right.

Comment 3: For the phonon calculation, the reviewer did not see any computational details, which is not acceptable. Also, the vdW should be concerned for phonon calculations. A thorough convergence test should be conducted for phonons as it is sensitive to computational parameters.

Reply to Comment 3: Thanks to the reviewer for the suggestion. Accordingly, we added the convergence tests and computational details to **Supplementary Note 1**. The phonon band structure of monolayer β - In_2Se_3 in our work was calculated using the finite displacement method with VASP and Phonopy codes. Firstly, we performed structural optimization calculations of the unit cell with different force convergence tolerance and k -point mesh grids. Secondly, the structures generated in the first step were expanded into a $6 \times 6 \times 1$ supercell to calculate the interatomic forces. Finally, the force constant and the phonon band structure were obtained. The results are shown in **Figures R7(a)-(d)** and the detailed computational parameters are shown in **Table R2**. As we can see from the convergence test, here with four sets of different calculation parameters, the phonon band dispersion exhibit negligible differences. Especially, the flat soft phonon modes along the Γ -M path, which induces the CDW orders, can be observed in all the cases, thereby validating the accuracy of the present phonon band structure.

Figure R7. Phonon band structures generated by different computational parameters for convergence tests, and the details are listed in **Table R2**.

Table R2. The detailed parameters of the phonon band structure calculations for the convergence test in **Figures R7(a-d)**.

Calculation	Force for geometry optimization	k-point mesh for geometry optimization (Unit cell)	k-point mesh for interatomic force (Supercell)
a in Fig. R7	10^{-6} eV/Å	13×13×1	1×1×1
b in Fig. R7	10^{-7} eV/Å	16×16×1	1×1×1
c in Fig. R7	10^{-8} eV/Å	21×21×1	1×1×1
d in Fig. R7	10^{-8} eV/Å	21×21×1	2×2×1

Additionally, phonon calculations considering D3 vdW interaction corrections were also conducted. The parameters for geometry optimization and interatomic force calculations were the same as those used for the phonon band structure calculations in **Figure R7(c)**. The results with and without D3 vdW corrections are shown in **Figure R8**. The phonon band structure, when considering the D3 vdW correction, remains almost the same and especially also exhibits the flat imaginary frequencies in the optical phonon branch along the Γ -M path. According to the suggestions of the referee, we have added the computational details and convergence test to **Note 1 of Supplementary Information**, the detailed content can be found in the **Reply to Comment 5**.

Figure R8. Phonon band structures of monolayer β - In_2Se_3 (a) with and (b) without D3 vdW interaction correction.

Detail changes in the main text (highlighted in red) include:

Page 17

The phonon band structure is calculated using the Phonopy code⁶⁷, the convergence tests are given in Supplementary Fig. 1. Further details about the calculations are provided in Note 1 of the Supplementary Information.

Comment 4: The substantial imaginary frequency of phonon band structure should be analyzed. The detailed eigen motions at the two negative modes at M and the degenerate one at G should be presented.

Reply to Comment 4: According to the suggestion, we further presented the four detailed eigen motions of the imaginary frequencies at the Γ and M points in **Figure R9**. As shown by the phonon band structure in **Figure R9(a)**, there are two degenerate negative modes marked as Modes 1 and 2 at the Γ point, the detailed eigen motions are also shown in **Figure R9(b)**. Both the modes are the vibrations of middle layer Se [Se(m)] atoms, where Mode 1 corresponds to the in-plane vibration of Se(m) atoms along the direction of lattice vector a , while Mode 2 represents the in-plane vibration of Se(m) atoms perpendicular to Mode 1. These two eigenmodes should be closely related to the ferroelectric phases of monolayer In_2Se_3 (i.e., FEA and FEB phases) mentioned in **Figure 2** of the

main text. For the lowest (imaginary-frequency) phonon branch along the Γ -M path, there are also eigen vibrations of Se(m) atoms along the a -direction, but with a phase difference. Taking Mode 3 at point M as an example in **Figure R9(c)**, the neighboring Se(m) atoms vibrate out of phase, which should be closely related to the anti-ferroelectric (i.e., AFE) phases shown in **Figure 2** of the main text. By the combination of the soft modes along the path Γ -M, different kinds of CDW orders, such as the c-SoD phase driven by the distortion of Se(m) atoms, therefore can be formed.

Figure R9. Eigen motion analysis of representative phonon modes with imaginary frequencies. (a) Phonon band structure of monolayer β_c -In₂Se₃. (b) The illustrations of the two degenerate phonon modes (Mode 1 and Mode 2) at the Γ point, marked by a red dot in (a). (c) The illustrations of the two phonon modes (Mode 3 and Mode 4) at the M point, marked by yellow and orange dots in (a), respectively. In (b) and (c), the upper row displays the side view, while the lower row shows the corresponding top view.

The Mode 4, marked by the orange dot in **Figure 9(a)**, involves the motions of all the atoms [not just Se(m) atoms], and it exhibits complex *rocking-chair-like* vibration behavior. The dynamic schematic of the eigen motion for mode 4 is provided in **Supplementary Video 1**. For a better understanding of the relationship between flat imaginary frequencies and multiple CDWs, we add the content of eigen motions of Modes 1, 2, and 3 into **Note 4 of Supplementary Information** and also add a discussion in the main text.

Detail changes in the main text (highlighted in red) include:

Page 9

From the perspective of the EPC effect, the formation of FE phases can be attributed to the imaginary phonon modes at the Γ point marked by the red dot in Fig. 2a. The eigen motions of these two degenerate modes involve the in-phase vibrations of Se(m) atoms in the in-plane direction, as detailed in the Supplementary Fig. 5 of Supplementary Note 4.

Page 9-10

Fig. 2e shows the AFE phase with opposite polarizations of adjacent units, and from the perspective of EPC effect, it corresponds to the opposite vibrations of adjacent Se(m) atoms. For the lowest (imaginary-frequency) phonon branch along the Γ -M path in Fig. 2a, there are also eigen vibrations of Se(m) atoms along the lattice vector a , but with a phase difference. For the mode marked by the red dot at point M, the neighboring Se(m) atoms vibrate out of phase, which may correspond to the formation of the AFE phases, the detailed eigen motions are presented in Supplementary Fig. 5 of

The three off-center primitive cells (\mathbf{A}_1 , \mathbf{A}_2 , and \mathbf{B}) that triggered by the significant imaginary frequencies together with the centrosymmetric cell [i.e., $\text{Se}(m)$ at the center of the Mexican-hat PES, named \mathbf{C}], can be served as the basic units of the CDW orders.

Comment 5: The DFT calculations are not well documented. This is essential for others to reproduce your results. This includes the details of MD simulations and electronic structure calculations. Was the HSE06 or HSE03 used for band structure calculations? What is the mixing parameters? Why did the authors choose HSE over metaGGA (e.g. TB-mBJ) for band gap evaluation?

Reply to Comment 5: According to the suggestion from the referee, we have added the detailed methods of the DFT calculations including electronic structure, MD, STM simulations, energy comparison, and phonon calculations to **Note 1 of Supplementary Information**, the specific content can be also seen as follows:

First-principles calculations

The Vienna *ab initio* simulation package (VASP) was used for DFT calculations¹. The projector augmented wave (PAW) pseudopotential was used to describe the electrons and the Perdew-Burke-Ernzerhof (PBE) exchange-correlation functional was employed^{2,3}. Since the usual PBE scheme is known to underestimate the bandgap, the Hyed-Scuseria-Ernzerhof (HSE06) hybrid functional was also used to calculate the band structure of β_c and *c*-SoD phases of monolayer $\text{III}_2\text{-VI}_3$ with the mixing parameters set to 25%⁴. The outmost *s* and *p* electrons for VI (VI=S, Se, Te), *s*, *p*, and *d* electrons for III (III=Al, Ga, In) are considered as valence states in the PAW potentials. The energy cut-offs of the plane wave basis sets were set as follows: 364 eV for Al_2S_3 , 312 eV for Al_2Se_3 , 312 eV for Al_2Te_3 , 367 eV for Ga_2S_3 , 367 eV for Ga_2Se_3 , 367 eV for Ga_2Te_3 , 364 eV for In_2S_3 , 311 eV for In_2Se_3 , 311 eV for In_2Te_3 . A vacuum layer with a thickness of 20 Å was constructed in the *z*-direction for all the cells in this study to weaken the effects of periodic boundary conditions. Electronic minimization was set to a tolerance of 10^{-6} eV, and ionic relaxation used a force tolerance of 0.01 eV/Å on each ion.

STM simulations

The simulated STM patterns were obtained using the VASP¹ package and the VASPKIT⁵ processing program. First, self-consistent calculations were performed using the VASP software package, and the wave function (WAVECAR) and charge density (CHGCAR) files were generated. Next, non-self-consistent calculations were performed by VASP to obtain the partial charge density file (PARCHG). Finally, the VASPKIT code was used to process the PARCHG file based on the Tersoff-Hamann approximation⁶ to generate STM images.

Energy comparison of various phases of 2D In_2Se_3

When comparing the relative energies of different phases of 2D In_2Se_3 , finer *k*-point meshes were used. For the geometry optimization, the tolerance of force on each ion was 0.01 eV/Å and the

energy cutoff was 311 eV, the k -point meshes were set as follows: $13 \times 13 \times 1$ for β_c and FE phases, $7 \times 13 \times 1$ for AFE_1 phase, $4 \times 13 \times 1$ for AFE_2 phase, $3 \times 13 \times 1$ for AFE_3 phase, $2 \times 13 \times 1$ for AFE_4 phase, $7 \times 9 \times 1$ for $2 \times \sqrt{3}$ phase, $5 \times 5 \times 1$ for $\sqrt{7} \times \sqrt{7}$ phase, $4 \times 4 \times 1$ for c -SoD phase, respectively. For static calculations, the k -point meshes were: $17 \times 17 \times 1$ for β_c and FE phases, $10 \times 17 \times 1$ for AFE_1 phase, $5 \times 17 \times 1$ for AFE_2 phase, $4 \times 17 \times 1$ for AFE_3 phase, $2 \times 17 \times 1$ for AFE_4 phase, $10 \times 13 \times 1$ for $2 \times \sqrt{3}$ phase, $7 \times 7 \times 1$ for $\sqrt{7} \times \sqrt{7}$ phase, $6 \times 6 \times 1$ for c -SoD phase, respectively.

Phonon calculations

The phonon band structure of monolayer β_c - In_2Se_3 in our work was calculated using the finite displacement method. The computational parameters of phonon band structure are rigorously tested for convergence, and the specific parameters and results are shown in Supplementary Fig. 1 (Figure R7) and Supplementary Table 1 (Table R2). Finally, the force convergence tolerance for geometry optimization was 10^{-8} eV/Å, and the k -point mesh was $21 \times 21 \times 1$ for the unit cell. The interatomic forces were computed using $6 \times 6 \times 1$ supercell and $1 \times 1 \times 1$ k -point mesh, with the tolerance for the convergence of total energy set to 10^{-8} eV. The force constant and the final phonon band structure were obtained using the Phonopy code⁷.

Molecular dynamics

The *ab initio* molecular dynamics (MD) simulations of monolayer c -SoD In_2Se_3 were performed using the VASP package. The NVT ensemble with Nosé thermostat was used⁸. A supercell contains 260 atoms was constructed for MD simulations and the k -point mesh is $1 \times 1 \times 1$. The time step is set to 1 fs, so the simulations were conducted for 10000 steps at 200 K.

In this study, indeed we have employed the high-level HSE06 hybrid functional to calculate the band structure for a better evaluation of band gap than that by the GGA functional, see Fig. 4i in the main text and **Supplementary Figures 14 and 16**. The mixing parameter is set to 25% for all the HSE06 calculations. Here, we use the HSE functional rather than the TB-MBJ functional for some reasons. First, the TB-MBJ calculation in 2D materials is possibly not as successful as that in 3D bulk materials [Phys. Rev. Lett. 102, 22, 226401 (2009); J. Chem. Theory Comput. 16, 4, 2654 – 2660 (2020); J. Phys. Chem. C 125, 20 11206 – 11215 (2021)]. Second, the official manual of the VASP code points out that the TB-MBJ calculations may suffer from slow convergence issues. Therefore, we chose HSE functional for electronic band structure calculations.

Reviewer #3 (Remarks to the Author):

General Comment: In this paper, the authors utilize first-principles calculations, STM simulations, and MD simulations to identify a new charge density wave (CDW) phase in 2D β - In_2Se_3 . They suggest that the CDW does not stem from strong electron-electron interactions but is instead due to the flat and extensive imaginary phonon frequencies and the Mexican-hat potential energy surface (PES) of the Se(m) atoms. If validated, these findings would be significant, offering fresh insights and new avenues for investigating the origins of CDWs. The manuscript is well-written overall. However, a critical shortcoming is the lack of experimental evidence for the new CDW phase, specifically the chiral star of David (c-SOD) in 2D In_2Se_3 (and other 2D $\text{III}_2\text{-VI}_3$ materials). The paper also fails to provide a plausible explanation for the absence of experimental observations. Additionally, there are several unclear aspects in the manuscript, as outlined below. If the authors can adequately address these issues, I would recommend publication in Nature Communications.

Reply to General Comment: We sincerely thank the referee for his/her positive evaluation of our work. Below we respond to the referee's comments one by one. Especially, the explanation for the provisional absence of experiential observations for the new CDW phase (c-SOD phase) and a proposed strategy for future experimental exploration are shown in detail later, see **Reply to Comment 3**.

Comment 1: What new physical properties are associated with the c-SOD phase?

Reply to Comment 1: The new physical properties of the c-SoD phase can be divided into the following several aspects.

(1) Firstly, the physical origin is new. Unlike traditional strong electron-correlation SoD CDW materials such as 1T-NbSe₂, 2D In_2Se_3 has a relatively simple electronic structure, with only III-s and VI-p orbitals contributing to the formation of the CDW order. Further, the parental phase of c-SoD In_2Se_3 , i.e. β - In_2Se_3 , exhibits relatively large and flat imaginary frequencies in the optical phonon branch across the Brillouin zone, which correspond to the various distortions of Se(m) atoms. By the combination of units with different Se(m) twists, multiple CDWs can be formed in 2D β - In_2Se_3 .

Figure R10. Electronic-property map of the pristine and CDW phases of 2D $\text{III}_2\text{-VI}_3$ (this work) and previously reported 2D or quasi-2D CDW materials¹⁻¹⁴. The horizontal axis indicates the bandgap of the high-symmetry phase, and the vertical axis indicates the bandgap of the CDW phase.

(2) Secondly, the present 2D c-SoD phase exhibits significantly larger band gap compared to other 2D CDWs, see **Figure R10**. The previously reported 2D materials typically exhibit only a slight band gap opening during CDW transitions. However, for the nine 2D $\text{III}_2\text{-VI}_3$ materials, their c-SoD phases all exhibit semiconducting behavior with a significant bandgap. Therefore, the CDW transitions in 2D $\text{III}_2\text{-VI}_3$ involve both metal-to-insulator and insulator-to-insulator transitions, which makes 2D $\text{III}_2\text{-VI}_3$ independent at the top of the CDW electronic-property map in **Figure R10**.

(3) Thirdly, the unique c-SoD configuration is expected to realize the atomic-scale ferroelectric vortex. The ferroelectric vortex can be regarded as an analog of magnetic vortex and ferroelastic vortex [Nano Lett. 21, 8, 3533 (2021)]. This disturbance-insensitive topology may avoid the "cross-talk" between adjacent information bits as much as possible, thereby possibly achieving high-density non-volatile memory [J. Am. Chem. Soc. 142, 52, 21932 (2020)]. As shown in **Figure R11(a)**, the c-SoD order is formed by the collective clockwise or counterclockwise twisting of six $\text{Se}(m2)$ atoms and the contraction of six $\text{Se}(m3)$ atoms toward the $\text{Se}(m1)$ atom. Although the c-SoD phase exhibits centrosymmetry and zero net polarization, there are dipoles in each unit caused by the distortion of $\text{Se}(m2 \text{ or } m3)$, see **Figure R11(b)**. In particular, for the six units containing $\text{Se}(m2)$ atoms, their electric dipoles exhibit a vortex arrangement, which will provide a very interesting platform for the study of ferroelectric vortex within the atomic scale.

Figure R11. Ferroelectric vortex in 2D c-SoD In_2Se_3 . (a) Schematic diagram of the distortion of $\text{Se}(m)$ atoms in the c-SoD phase, with $\text{Se}(m2)$ atoms highlighted by red circles. (b) Electric dipole caused by the off-center distortion of $\text{Se}(m2)$ atom. (c) The vortex arrangement of electric dipoles caused by the $\text{Se}(m2)$ atoms.

References in Fig. R10 are listed as below.

1. Liu L, et al. Direct identification of Mott Hubbard band pattern beyond charge density wave superlattice in monolayer 1T-NbSe₂. Nat. Commun. 12, 1978 (2021).
2. Jiang T, et al. Two-dimensional charge density waves in TaX₂ (X=S,Se,Te) from first principles. Phys Rev B 104, 075147 (2021).
3. Tresca C, Calandra M. Charge density wave and spin 1/2 insulating state in single layer 1T-NbS₂. 2D Mater 6, 035041 (2019).
4. Lin H, et al. Scanning tunneling spectroscopic study of monolayer 1T-TaS₂ and 1T-TaSe₂. Nano Res 13, 133-137 (2020).
5. Kolekar S, Bonilla M, Ma Y, Diaz HC, Batzill M. Layer- and substrate-dependent charge density wave criticality in 1T-TiSe₂. 2D Mater 5, 015006 (2018).
6. Chen G, et al. Correlating structural, electronic, and magnetic properties of epitaxial VSe₂ thin films. Phys Rev B 102, 115149 (2020).
7. Zhang D, et al. Strain engineering a $4a \times \sqrt{3}a$ charge-density-wave phase in transition-metal dichalcogenide 1T-VSe₂. Phys Rev Mater 1, 024005 (2017).
8. Liu M, Wu C, Liu Z, Wang Z, Yao D-X, Zhong D. Multimorphism and gap opening of charge-density-wave phases in monolayer VTe₂. Nano Res 13, 1733-1738 (2020).
9. Tan H, Liu Y, Wang Z, Yan B. Charge Density Waves and Electronic Properties of Superconducting Kagome Metals. Phys Rev Lett 127, 046401 (2021).
10. Calandra M, Mazin II, Mauri F. Effect of dimensionality on the charge-density wave in few-layer 2H-NbSe₂. Phys Rev B 80, 241108 (2009).
11. Chen P, et al. Unique Gap Structure and Symmetry of the Charge Density Wave in Single-Layer VSe₂. Phys Rev Lett 121, 196402 (2018).
12. Miller DC, Mahanti SD, Duxbury PM. Charge density wave states in tantalum dichalcogenides. Phys Rev B 97, 045133 (2018).
13. Yang Y, et al. Enhanced superconductivity upon weakening of charge density wave transport in 2H-TaS₂ in the two-dimensional limit. Phys Rev B 98, 035203 (2018).
14. Ryu H, et al. Persistent Charge-Density-Wave Order in Single-Layer TaSe₂. Nano Lett 18, 689-694 (2018).

Comment 2: The authors selected In₂Se₃, which is known for its complex phase transition behavior. What are the main differences among 2D III₂-VI₃ materials? Are similar CDW phases, such as the c-SOD phases, found in other 2D III₂-VI₃ materials?

Reply to Comment 2: We thank the referee very much for this insightful comment. The main difference between 2D In₂Se₃ and other 2D III₂-VI₃ materials should be the electronic band gap. **Figure R12** displays the HSE06 band structure of nine 2D β_c III₂-VI₃ materials. The corresponding band gaps are also listed in **Table R3**. The band gaps are 1.06 eV for In₂Se₃ and 0 ~ 2.38 eV for the other eight III₂-VI₃ materials.

Figure R12. HSE06 band structures of nine 2D β_c III₂-VI₃. (a) Al₂S₃, (b) Al₂Se₃, (c) Al₂Te₃, (d) Ga₂S₃, (e) Ga₂Se₃, (f) Ga₂Te₃, (g) In₂S₃, (h) In₂Se₃, and (i) In₂Te₃.

Table R3. HSE06 band gaps of nine 2D β_c III₂-VI₃.

	S	Se	Te
Al	2.38 eV	1.40 eV	0.21 eV
Ga	1.26 eV	0.41 eV	0 eV
In	1.90 eV	1.06 eV	0.19 eV

According to our study, other eight 2D β_c III₂-VI₃ materials also exhibit similar Mexican-hat potential energy surface (PES) as β_c -In₂Se₃ does. This indicates that other 2D III₂-VI₃ materials may also exhibit similar CDW orders as 2D In₂Se₃ has. To address the question regarding the universality of CDW in 2D III₂-VI₃ materials, we conducted a series of geometric optimizations. As we anticipated, the c-SoD CDW can also form in other eight 2D III₂-VI₃ materials, and their atomic structures as well as that of In₂Se₃ are shown in **Figure R13**. The corresponding HSE06 band structures of the nine c-SoD materials are also shown in **Figure R14**. Compared with the pristine β_c phases, the band gaps of c-SoD phases (0.91 ~ 3.52 eV) all exhibit a significant increase (listed in **Table R4**). These results demonstrate the generality of the c-SoD CDW in the 2D III₂-VI₃ family.

Figure R13. Atomic structures of the c-SoD CDW phase in 2D III₂-VI₃. (a) Al₂S₃, (b) Al₂Se₃, (c) Al₂Te₃, (d) Ga₂S₃, (e) Ga₂Se₃, (f) Ga₂Te₃, (g) In₂S₃, (h) In₂Se₃, and (i) In₂Te₃.

Figure R14. HSE06 band structures of the c-SoD CDW phase of 2D III₂-VI₃. (a) Al₂S₃, (b) Al₂Se₃, (c) Al₂Te₃, (d) Ga₂S₃, (e) Ga₂Se₃, (f) Ga₂Te₃, (g) In₂S₃, (h) In₂Se₃, and (i) In₂Te₃.

Table R4. HSE06 band gaps of nine 2D c-SoD III₂-VI₃.

	S	Se	Te
Al	3.52 eV	2.62 eV	1.51 eV
Ga	2.65 eV	1.78 eV	0.91 eV
In	2.79 eV	2.09 eV	1.26 eV

Comment 3: Given the slight relaxed energy difference between the $\sqrt{13}\times\sqrt{13}$ and $2\times\sqrt{3}$ phases (see Fig. 2g), why has the $\sqrt{13}\times\sqrt{13}$ phase not been observed experimentally, while the $2\times\sqrt{3}$ phase has?

Reply to Comment 3: We thank the referee for this insightful question. In 2019 Zhang et al. first experimentally observed the $2\times\sqrt{3}$ phase by cooling the β - In_2Se_3 sample from room temperature to 77 K [ACS Nano 13, 7, 8004–8011 (2019)]. In their work, the β phase corresponds to the one whose middle layer Se [Se(*m*)] atoms uniformly deviate from the central position, i.e., the in-plane ferroelectric phase. Similarly, the $2\times\sqrt{3}$ phase was also reported to exhibit in-plane ferroelectricity [ACS Nano. 13, 7, 8004–8011 (2019), Nat. Commun. 15, 718 (2024)]. In other words, the presence of in-plane ferroelectricity in both phases (which can be regarded as a kind of “seed”) may facilitate the transition to the $2\times\sqrt{3}$ phase. However, for the $\sqrt{13}\times\sqrt{13}$ c-SoD phase, the distortions of Se(*m*) atoms are complicated and multi-directional, i.e., Se(*m*2) twists around Se(*m*1) either clockwise or counterclockwise while Se(*m*3) contracts towards Se(*m*1), and the net electric dipole moment of the $\sqrt{13}\times\sqrt{13}$ phase is zero. Therefore, lack of in-plane (net) ferroelectricity in the $\sqrt{13}\times\sqrt{13}$ phase should be an important reason for still not observing the $\sqrt{13}\times\sqrt{13}$ phase in previous experiments mentioned above.

Figure R15. Two methods including optically electronic excitation and electron doping proposed to eliminate in-plane ferroelectricity in the initial β phase and a proof-of-concept validation by DFT geometry optimization. (a) 1% (optical) excitation of valence electrons from the valence band into the conduction band. (b) Directly doping an amount of 1% of the total valence electrons into the conduction band. The structures of the initial and final phases are shown for the two methods.

Furthermore, the analysis above inspires us to propose a potential strategy for obtaining the $\sqrt{13}\times\sqrt{13}$ c-SoD phase, which may aid future experimental exploration. By first-principles calculations, we carry out the discussion as below. Due to the “zero” net dipole moment of the $\sqrt{13}\times\sqrt{13}$ c-SoD phase, a feasible approach to obtaining the c-SoD phase is to eliminate the in-plane ferroelectric domains (as seeds of forming the $2\times\sqrt{3}$ phase) in the parent material before lowering the temperature. As shown in the experiment [ACS Nano 13, 7, 8004–8011 (2019)], at

room temperature the initial phase of β - In_2Se_3 is an in-plane ferroelectric phase with middle layer Se [$\text{Se}(m)$] atoms uniformly deviate from the central position. Here, first of all, we suggest using optically electronic excitation (for example, by ultrafast laser pumping) or electron doping to eliminate the in-plane ferroelectricity. Because the photoexcitation-driven PES modification has been proven to be an effective method for structural manipulation [Phys. Rev. Lett. 120, 18, 185701 (2018)]. **Figure R15** demonstrates that 1% electronic excitation or 1% electron doping can modulate the original Mexican-hat PES into a single-well-like PES and then the β - In_2Se_3 is successfully transformed from an in-plane ferroelectric phase to a paraelectric β_c phase, see the final relaxed structures. Second, the β_c - In_2Se_3 phase is cooled down to a low temperature to form the $\sqrt{13}\times\sqrt{13}$ c-SoD phase. In order to test the idea, we performed a low-temperature molecular dynamics (MD) starting from the paraelectric β_c phase, where the temperature is set to ≤ 60 K. As shown in **Figure R16**, at 0.5 ps the pristine high-symmetry β_c phase really transforms into the $\sqrt{13}\times\sqrt{13}$ c-SoD phase. The dynamic process of the phase transition at low temperature is also provided in **Supplementary Video 2**.

Figure R16. Transient structure during the low-temperature MD simulation of transition from the paraelectric β_c phase to the $\sqrt{13}\times\sqrt{13}$ c-SoD phase.

In the revised manuscript, we added a discussion to explain the reason for still not experimentally observing the $\sqrt{13}\times\sqrt{13}$ phase. Furthermore, we provided a potential strategy for obtaining the $\sqrt{13}\times\sqrt{13}$ c-SoD phase, which may aid future experimental exploration.

Detail changes in the main text (highlighted in red) include:

Page 11

In addition, the energies of the $2\times\sqrt{3}$ phase and the c-SoD phase are very close to each other. Considering that the $2\times\sqrt{3}$ CDW phase has been experimentally confirmed to be stable at 170 K^{44} . Therefore, we predict that the c-SoD phase may also be experimentally stable in a similar

temperature range, which is also consistent with the stability deduced from the MD in **Supplementary Figure 4**. The current experimental observations of the in-plane polarized $2 \times \sqrt{3}$ phase are achieved by cooling the FE β phase, which also has an in-plane ferroelectricity as the seed for the transition. However, to obtain the centrosymmetric *c*-SoD phase (with zero net electric dipole moment) in experiments, we suggest one should eliminate the ferroelectric seed in the FE β phase before cooling the temperature, possibly with effective methods like optically electronic excitation or electron doping⁴⁵.

Comment 4: In Fig. 1c, the labels for left-hand and right-hand are difficult to distinguish. I suggest using different colors and/or line types to make them more distinguishable.

Reply to Comment 4: Thank you very much for the suggestion. Accordingly, we changed the color of the left-hand and right-hand labels to orange and green respectively, as shown in **Figure R17(c)**.

Figure R17. The revised Fig. 1c with color modifications for the left-hand and right-hand labels.

Finally, we once again sincerely thank all the referees for their many efforts on our manuscript.

REVIEWERS' COMMENTS

Reviewer #1 (Remarks to the Author):

I have carefully read the revised manuscript along with its supplemental information, and the responses to all three reviewers. I think the authors have carefully addressed all the comments from the theoretical side. Note that even though some experimental observations may still lack, such a careful theoretical prediction is reliable and could stimulate further experimental verifications. One recent example is the anti-DoS CDW in 135 kagome superconductors, which has received quick experimental demonstrations after theoretical prediction. Considering the importance of chiral physics in recent years, I would recommend its publication as is.

Reviewer #2 (Remarks to the Author):

The authors have taken a serious effort on addressing the reviewer's comment. I think that the paper can be accepted for publication now.

Reviewer #3 (Remarks to the Author):

The authors have satisfactorily addressed my comments and concerns in their response and the revised manuscript. I believe the manuscript is now suitable for publication in Nature Communications.

Point-by-Point Response to Reviewers' Comments

Reviewer #1 (Remarks to the Author):

I have carefully read the revised manuscript along with its supplemental information, and the responses to all three reviewers. I think the authors have carefully addressed all the comments from the theoretical side. Note that even though some experimental observations may still lack, such a careful theoretical prediction is reliable and could stimulate further experimental verifications. One recent example is the anti-DoS CDW in 135 kagome superconductors, which has received quick experimental demonstrations after theoretical prediction. Considering the importance of chiral physics in recent years, I would recommend its publication as is.

Reply to Comment: We are very grateful to the reviewer for his/her support in publishing our manuscript as well as the thoughtful comments regarding both the theoretical aspects and the potential for future experimental verification.

Reviewer #2 (Remarks to the Author):

The authors have taken a serious effort on addressing the reviewer's comment. I think that the paper can be accepted for publication now.

Reply to Comment: We are very grateful to the reviewer for agreeing to accept the manuscript. We thank the reviewer for their valuable time and thoughtful evaluation during the review process.

Reviewer #3 (Remarks to the Author):

The authors have satisfactorily addressed my comments and concerns in their response and the revised manuscript. I believe the manuscript is now suitable for publication in Nature Communications.

Reply to Comment: We sincerely appreciate the reviewer's positive feedback and support for the publication of our work.